

# Cryptic diversity and phylogeographic patterns of *Deto echinata* (Isopoda: Detonidae) in southern Africa

Carlos A. Santamaria[1] and Charles L. Griffiths[2]

[1] Department of Biology, The University of Tampa, Tampa, FL, United States of America
[2] Department of Biological Sciences and Marine Research Institute, University of Cape Town, Rondebosch, South Africa

## ABSTRACT

Recent phylogeographic studies of poorly-dispersing coastal invertebrates in highly biodiverse regions have led to the discovery of high levels of cryptic diversity and complex phylogeographic patterns that suggest isolation, geological, and ecological processes have shaped their biodiversity. Studies of southern African coastal invertebrates have uncovered cryptic diversity for various taxa and phylogeographic patterns that, although sharing some similarities across taxa, do differ. These findings underscore the need for additional studies to better understand the biodiversity levels, distributional patterns, and processes responsible for producing coastal biodiversity in that region. The coastal isopod *Deto echinata* is of particular interest, as its complex taxonomic history, poor dispersal capabilities, and broad geographic distribution suggest the potential for cryptic diversity. We use mitochondrial and nuclear sequences to characterize *D. echinata* individuals from localities ranging from northern Namibia to Glentana, about 2,500 km along the coastline on the south coast of South Africa. These are used to assess whether *D. echinata* harbors cryptic genetic diversity and whether phylogeographic distributional patterns correlate with those previously documented for other coastal isopods in the region. Analysis of mitochondrial and nuclear sequences revealed two deeply-divergent lineages that exhibit a distributional break in the Cape Peninsula region. These findings suggest *D. echinata* is a cryptic species complex in need of taxonomic revision and highlight the need for further taxonomic and phylogeographic studies of similarly poorly-dispersing coastal invertebrates in southern Africa.

Corresponding author
Carlos A. Santamaria,
santamaria.carlos.a@gmail.com

## INTRODUCTION

Recent phylogeographic studies of poorly-dispersing coastal invertebrate species have shown that these taxa often harbor high levels of cryptic diversity, as well as complex phylogeographic patterns (*Taiti et al., 2003*; *Chan, Tsang & Chu, 2007*; *Jung et al., 2008*; *Varela & Haye, 2012*; *Xavier et al., 2012*; *Eberl et al., 2013*; *Santamaria et al., 2013*; *Santamaria, Mateos & Hurtado, 2014*; *Santamaria et al., 2017*; *Greenan, Griffiths & Santamaria, 2018*; *Hurtado et al., 2018*; *Santamaria, 2019*; *Santamaria & Koch, 2023*). Phylogeographic studies of coastal invertebrates in South Africa are no exception and have led to the discoveries of deeply-divergent lineages and contrasting phylogeographic patterns

(*Evans et al., 2004*; *Zardi et al., 2007*; *Reynolds, Matthee & Heyden, 2014*; *Greenan, Griffiths & Santamaria, 2018*; *Mbongwa et al., 2019*; *Simon et al., 2020*; *von der Heyden, Mbongwa & Hui, 2020*; *Bezuidenhout et al., 2021*). For instance, while several coastal invertebrates have been shown to exhibit strong phylogeographic breaks in the region around the Cape of Good Hope (*e.g.*, *Teske et al., 2006*; *Greenan, Griffiths & Santamaria, 2018*; *von der Heyden, Mbongwa & Hui, 2020*), a major biogeographic transition zone (*Griffiths et al., 2010*), unique phylogeographic breaks not shared amongst taxa have also been reported (*Teske et al., 2006*; *Greenan, Griffiths & Santamaria, 2018*; *Mbongwa et al., 2019*; *Bezuidenhout et al., 2021*). The discovery of highly divergent genetic lineages have led to suggestions that some species in the region may represent cryptic species complexes and thus cryptic diversity (*Greenan, Griffiths & Santamaria, 2018*). These findings have furthered our understanding of the biodiversity of the region and suggest that other poorly dispersing organisms also harbor previously unreported cryptic diversity and unique phylogeographic patterns. Molecular studies of such taxa may thus be informative on the biodiversity of southern Africa and on the processes driving diversification along coastlines in the region.

Previous molecular studies of coastal isopods with complex taxonomic histories have shown them to harbor cryptic diversity and complex phylogenetic patterns (*Hurtado, Mateos & Santamaria, 2010*; *Hurtado, Lee & Mateos, 2013*; *Santamaria et al., 2013*; *Santamaria, Mateos & Hurtado, 2014*; *Santamaria & Koch, 2023*). Drift-line isopods in the genus *Deto* (*Guérin-Méneville, 1836*) are one potential taxonomic group of interest. This genus is currently considered to have four well-established species, *D. aucklandiae* Thomson 1879 and *D. bucculenta* Nicolet 1849 from New Zealand and its outlying islands; *D. marina* Chilton 1885 from southern Australia; and *D. echinata* (*Guérin-Méneville, 1836*) from southern Africa (with an outlying population reported from St Paul Island in the Indian Ocean) (*Schmalfuss, 2003*). A fifth species, *D. whitei* (*Kinahan, 1859*), is of doubtful status, as its type location remains unknown and it has not been reported since its original and extremely cursory description by *Kinahan (1859)*. This has led several authors (*Budde-Lund, 1885*; *Chilton, 1915*; *Schmalfuss, 2003*) to suggest that it is a synonym of *D. echinata*; indeed, the original author himself remarked on its close similarly to that species.

*Deto echinata* was first described by *Guérin-Méneville (1836)* and is readily distinguished by the pairs of long dorsal 'horns' projecting from each thoracic segment. The type locality of the species remains obscure, since the original description simply gives its type locality as 'd'Orient' or 'in the east' (*Guérin-Méneville, 1836*). Moreover, the original specimens cannot be traced. The subsequent taxonomic history of *D. echinata* is long and confused. *Krauss (1843)* provided the first unambiguous South African record, which he collected from the sea-shore in Table Bay. *Budde-Lund (1885)* also recorded *D. echinata* from the region, as well as another species, *D. acinosa,* which he described as new. *Chilton (1915)* provided a full review of the genus as it stood at that time and retained both South African species, which he distinguished primarily by the sizes of the dorsal projections, which he described as long in *D. echinata*, but as reduced to tubercles in *D. acinosa*. The same distinction was maintained by *Panning (1924)* with respect to specimens from Namibia, but *Barnard (1932)* concluded that *D. acinosa* simply represented 'not fully grown, but not necessarily sexually immature' specimens and amalgamated all southern African specimens under the

name *D. echinata* (although he still depicted the *acinosa* 'form'). Subsequent authors have referred to the species simply as *D. echinata* and have not recognized sub-specific taxa (*Kensley, 1978*; *Coleman & Leistikow, 2001*; *Branch et al., 2016*). This decision appears to be supported by *Glazier, Clusella-Trullas & Terblanche (2016)* who showed that horn length in this species not only varies with sex and body size, but is also strongly influenced by nutritional condition.

An outlying population of *D. echinata* is also known from Saint Paul Island, in the Southern Indian Ocean (33°S, 77°E), midway between South Africa and Australia. That population was originally described as a separate species, *D. armata*, by *Budde-Lund (1906)*. *Chilton (1915)* continued to recognize this as a separate species, although noting its similarity to *D. echinata*. However, both *Barnard (1932)* and *Coleman & Leistikow (2001)* regarded it as a junior synonym of *D. echinata* and that remains its current status.

*Deto echinata* is an air-breathing species found in vast numbers in the upper intertidal and supratidal regions of rocky shores and has a reported range from northern Namibia to about Knysna on the south coast of South Africa (Fig. 1, *Barnard, 1932*; *Kensley, 1978*). It occurs along with similar-looking (but spineless and only distantly-related), members of the genus *Ligia* (family Ligiidae), all species feeding on drift algae deposited along the drift-line of rocky shores. Despite the extraordinary abundance of *D. echinata* at some sites, its complex taxonomic history, and the range overlap with *Ligia* spp., for which cryptic diversity has previously been reported (see *Greenan, Griffiths & Santamaria, 2018*), there have been no previous publications on phylogeography or cryptic diversity of this isopod.

In this study, we use mitochondrial and nuclear markers to characterize *D. echinata* individuals from diverse localities across its range in Namibia and South Africa to determine whether cryptic genetic diversity exists within these. We also examine the distributional patterns of the genetic diversity of *D. echinata* and compare them to previous phylogeographic patterns for coastal isopods in the region.

## MATERIALS AND METHODS

### Specimen collections

*Deto* individuals were hand-collected from the upper intertidal zone at 16 coastal localities chosen as to span the known geographic range of *Deto echinata* as well as reported phylogeographic breaks in the region (Fig. 1). Localities span from Rocky Point in northern Namibia to Glentana, on the south coast of South Africa, a coastal distance of ~2,500 km. Sampling attempts in some other localities in the southern coastline of South Africa east of L'Agulhas (*e.g.*, Arniston, De Hoop, and Mossel Bay) were not successful, as no specimens could be found, despite suitable habitat being searched. Detailed information on sampling localities included in this study is provided in Table 1. All samples were field-preserved and stored in 95% ethanol pending molecular analyses. Field collections were carried out under Scientific Collection Permit RES2017/53 issued to CG jointly by the South African Departments of Environmental Affairs and of Agriculture, Forestry and Fisheries.

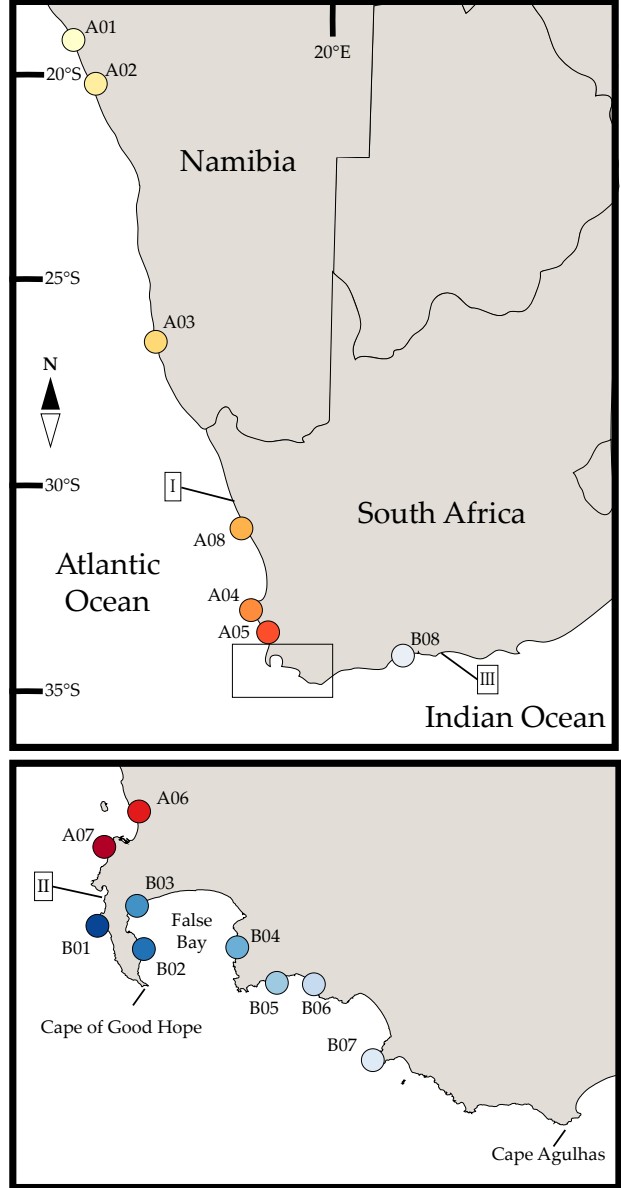

**Figure 1  Sampled localities across Namibia and South Africa.** Sampled localities were: A01-Rocky Point, Namibia; A02-Moewe Bay, Namibia; A03-Luderitz, Namibia; A04-Jacobsbaai, South Africa; A05-Ganzekraal, South Africa; A06-Blauberg, South Africa; A07-Bakoven, South Africa; A08-Island Wreck, South Africa; B01-Kommetjie, South Africa; B02-Simon's Town, South Africa; B03-Kalk Bay, South Africa; B04-Koelbaai, South Africa; B05-Kleinmond, South Africa; B06-Onrus, South Africa; B07-Gansbaai, South Africa; B08-Glentana, South Africa. Colors and labels correspond with other figures. Roman numerals indicate phylogeographic breaks previously reported in other coastal invertebrates from the region: (I) region between Hondeklip Bay and Kleinzee (*Mbongwa et al., 2019*); (II) region between Ganzekraal and Kommetjie (*Greenan, Griffiths & Santamaria, 2018*); (III) region between L'Agulhas and the Gourits Estuary (*Teske et al., 2006*). Map is edited from a public domain map produced by Lokal_Profil. Original vector map is available at https://commons.wikimedia.org/wiki/File:BlankMap-Africa.svg.

**Table 1 Localities included and corresponding GenBank Accession Numbers for all genetic markers used, latitude, and longitude.** Map labels correspond with all other figures and tables.

| Location | Map Label | N inds | Latitude | Longitude | Acc. No. COI | Acc. No. 16S rDNA | Acc. No. 12S rDNA | Acc. No. 28S rDNA |
|---|---|---|---|---|---|---|---|---|
| Rocky Point, Namibia | A01 | 5 | 18°58′59″S | 12°28′59″E | OQ852635 OQ852636 OQ852637 | OQ870101 OQ870102 OQ870103 OQ870104 | OQ870136 OQ870137 | OQ870153 |
| Moewe Bay, Namibia | A02 | 5 | 19°22′25″S | 12°42′15″E | OQ852634 | OQ870100 | OQ870135 | N/A |
| Luderitz, Namibia | A03 | 2 | 26°39′47″S | 15°04′55″E | OQ852632 OQ852633 | N/A | N/A | N/A |
| Jacobsbaai, South Africa | A04 | 5 | 32°58′26″S | 17°53′06″E | OQ852638 OQ852639 OQ852640 | OQ870105 | OQ870138 OQ870139 | N/A |
| Ganzekraal, South Africa | A05 | 5 | 33°31′18″S | 18°19′19″E | OQ852641 OQ852642 OQ852643 | OQ870106 | OQ870140 OQ870141 | OQ870154 OQ870155 |
| Blauberg, South Africa | A06 | 5 | 33°48′26″S | 18°28′37″E | OQ852675 OQ852676 OQ852677 | OQ870129 OQ870130 OQ870131 | N/A | N/A |
| Bakoven, South Africa | A07 | 4 | 33°57′54″S | 18°22′25″E | OQ852667 OQ852668 OQ852669 | OQ870127 | N/A | N/A |
| Island Wreck, South Africa | A08 | 5 | 30°55′1″S | 17°36′12″E | OQ852670 OQ852671 OQ852672 OQ852673 OQ852674 | OQ870128 | OQ870152 | N/A |
| Kommetjie, South Africa | B01 | 5 | 34°08′17″S | 18°19′24″E | OQ852659 OQ852660 OQ852661 | OQ870117 | N/A | OQ870157 OQ870158 |
| Simon's Town, South Africa | B02 | 5 | 34°11′0″S | 18°26′0″E | OQ852662 OQ852663 OQ852664 | OQ870118 OQ870119 OQ870120 OQ870121 OQ870122 | OQ870148 OQ870149 | OQ870159 |
| Kalk Bay, South Africa | B03 | 5 | 34°07′00″S | 18°26′00″E | OQ852665 OQ852666 | OQ870123 OQ870124 OQ870125 OQ870126 | OQ870150 OQ870151 | OQ870160 |
| Koelbaai, South Africa | B04 | 5 | 34°14′51″S | 18°51′15″E | OQ852644 OQ852645 | OQ870107 OQ870108 OQ870109 | OQ870142 OQ870143 | N/A |
| Kleinmond, South Africa | B05 | 5 | 34°20′22″S | 19°02′03″E | OQ852655 OQ852656 OQ852657 OQ852658 | OQ870114 OQ870115 OQ870116 | OQ870147 | N/A |

**Table 1** (*continued*)

| Location | Map Label | N inds | Latitude | Longitude | Acc. No. COI | Acc. No. 16S rDNA | Acc. No. 12S rDNA | Acc. No. 28S rDNA |
|---|---|---|---|---|---|---|---|---|
| Onrus, South Africa | B06 | 5 | 34°25′13″S | 19°10′35″E | OQ852650 OQ852651 OQ852652 OQ852653 OQ852654 | OQ870111 OQ870112 OQ870113 | OQ870146 | N/A |
| Gansbaai, South Africa | B07 | 5 | 34°35′10″S | 19°20′34″E | OQ852646 OQ852647 OQ852648 OQ852649 | OQ870110 | OQ870144 OQ870145 | OQ870156 |
| Glentana, South Africa | B08 | 5 | 34°02′60″S | 22°19′00″E | OQ852678 OQ852679 OQ852680 | OQ870132 OQ870133 OQ870134 | N/A | N/A |

## Molecular laboratory methods

Total genomic DNA was extracted from pereopods for 1–5 *Deto* individuals per locality using the Quick g-DNA MiniPrep Kit (Zymo Research, Irvine, CA, USA), following standard protocol instructions for animal tissues. For each specimen, we attempted to PCR-amplify three mitochondrial and one nuclear gene fragments, using previously published primers and conditions: (a) a 710-bp fragment of the *Cytochrome Oxidase I* (*COI*) mitochondrial gene using the LCO-1490 and HCO-2198 primers (*Folmer et al., 1994*); (b) a ~490-bp fragment of the *16S rDNA* mitochondrial gene using primers 16Sar and 16Sbr (*Palumbi, 1996*); (c) a ~495-bp of *12S rDNA* mitochondrial gene using primers crust-12Sf and crust-12Sr (*Podsiadlowski & Bartolomaeus, 2005*); and (d) a ~600-bp region of the *28S rDNA* gene using primers 28SA/28SB (*Whiting, 2002*). PCR products were checked for successful amplification using 1% agarose gels stained using SYBR Safe (Invitrogen, Carlsbad, CA, USA), with positive PCR amplicons sequenced at the Arizona Genetics Core (AZGC). Sequences were assembled, edited (*i.e.,* had primers removed), and inspected for evidence indicative of pseudogenes (*e.g.,* premature stop codons or frame shifts in the protein coding *COI* alignment), heteroplasmy (*e.g.,* multiple peaks in chromatograms of mitochondrial genes), and/or heterozygosity (*e.g.,* multiple peaks in chromatograms of the nuclear *28S rDNA*) using Geneious v8.1.9 (https://www.geneious.com/). No evidence of pseudogenes, heteroplasmy, nor heterozygous individuals was observed.

## Sequence alignments, phylogenetic analyses, and estimation of molecular divergence

Sequences produced in this study were sorted by gene, with each dataset then aligned independently using the MAFFT algorithm (*Katoh & Standley, 2013*; *Katoh, Rozewicki & Yamada, 2019*) as implemented in the GUIDANCE2 Server (*Sela et al., 2015*) with all settings as default except the number of bootstrap replicates, which was set to 100. Positions with an alignment score <1.00 in the final alignment were considered poorly aligned and excluded from posterior analyses. Mitochondrial genes were concatenated into a single alignment in SequenceMatrix v1.6.7 (*Vaidya, Lohman & Meier, 2011*). The nuclear *28S rDNA* gene was not incorporated into this dataset, given the low levels of variation

observed (see Results). Instead, we estimated relationships between *28S* haplotypes using the cladogram estimation algorithm of (*Templeton, Crandall & Sing, 1992*), as implemented by PopART v1.7 (*Leigh & Bryant, 2015*).

We used jModeltest v2.1.1 (*Darriba et al, 2012*) to determine the most appropriate model of DNA substitution for each mitochondrial gene fragment and the mitochondrial concatenated dataset by evaluating likelihood scores of 1,624 candidate models on a fixed BioNJ-JC tree, under the Akaike Information Criterion (AIC), corrected AIC (AICc), and the Bayesian Information Criterion (BIC). Selected models were used in phylogenetic reconstructions, unless the chosen model was not available in the software being used, or if the chosen model called for the joint estimation of Γ and I parameters. In case of the former, we used the next more complex model available in the software, while in the case of the latter, we used the simpler Γ.

We attempted to identify an appropriate outgroup to root our phylogenetic reconstructions by incorporating publicly available sequences for other *Deto* species; however, the only available data at the time of analyses in May of 2023 were *COI* sequences for *D. marina* (GenBank accession numbers: KR424585, KR424586, and EU364625). Thus, we aligned these sequences with the *D. echinata COI* sequences produced herein and conducted preliminary phylogenetic reconstructions in RAxML v8.2.12 (*Stamatakis, 2014*), as described below. The reconstructions recovered a split between two moderately supported *D. echinata* clades when rooted with *D. marina* (BS >75) with the topology of the tree remaining unchanged upon using a mid-point root. Thus, all posterior phylogenetic analyses were rooted using each identified clade to root the other clade as implemented by both *Mateos et al. (2012)* and *Santamaria & Koch (2023)*.

Phylogenetic reconstructions were carried out on the concatenated mitochondrial dataset using both Maximum Likelihood (ML) and Bayesian Inference (BI) under two partitioning schemes (*i.e.*, unpartitioned, partitioned by gene). ML searches were carried out in RAxML v8.2.12 (*Stamatakis, 2014*) and consisted of 1,000 bootstrap replicates followed by a thorough ML search under the GTR + Γ model run under the Rapid Bootstrap Algorithm (*Stamatakis, Hoover & Rougemont, 2008*) with all other settings as default. For each search, we estimated a majority-rule consensus of all bootstrap replicates using the SumTrees command of DendroPy v4.1.0 (*Sukumaran & Holder, 2010*). BI searches were carried out in MrBayes v3.2.5 (*Ronquist & Huelsenbeck, 2003*) and Phycas v2.2.0 (*Lewis, Holder & Swofford , 2015*). Searches in MrBayes consisted of two simultaneous searches of four chains run for $20 \times 10^6$ generations sampled every 5,000th generation, while Phycas searches consisted of a single search of $2 \times 10^6$ generations sampled every 50th generation. We determined whether Bayesian analyses had reached convergence if the average standard deviation of the split frequencies of independent runs was stable and close to zero, and if the Effective Sample Size (ESS) for posterior probabilities exceeded 200 when evaluated in Tracer v.1.7 (*Rambaut et al., 2018*). We estimated node support values by discarding all samples prior to stationarity (10–25% of sampled trees) and calculating a majority-rule consensus tree using the SumTrees command of DendroPy v4.1.0 (*Sukumaran & Holder, 2010*).

Lastly, Kimura-2-Parameter (K2P) pairwise genetic distances for the *COI* gene dataset were estimated using MEGA v11.0.13 (*Tamura, Stecher & Kumar, 2021*).

## Molecular species delimitation analyses

We determined the most likely number of putative species present in our *D. echinata* dataset using both distance (ASAP; *Puillandre, Brouillet & Achaz, 2021*) and tree-based (GMYC: *Fujisawa & Barraclough, 2013*, PTP: *Zhang et al., 2013*) molecular species delimitation analyses (hereafter MSDAs).

ASAP analyses were carried out separately on the *COI* dataset and on the concatenated mitochondrial dataset using the ASAP online server (https://bioinfo.mnhn.fr/abi/public/asap/) using the Kimura 2-Parameter (K2P) nucleotide evolution model, a ts/tv ratio of 2, and all other settings as default. Individuals in the concatenated mitochondrial dataset with missing data were removed prior to analyses.

PTP analyses were conducted on the consensus trees produced by phylogenetic reconstructions carried out in RAxML and MrBayes. PTP analyses were carried out under the Maximum Likelihood implementation of PTP (*Zhang et al., 2013*) using 500,000 MCMC iterations, a random seed, a burn-in of 0.10, and a thinning value of 100. GMYC analyses were conducted on ultrametric trees inferred in BEAST v2.1.3 (*Bouckaert et al., 2014*) using three different approaches: (a) assuming a constant rate of evolution and speciation under a Yule process (*Yule, 1925*; hereafter Constant + Yule, *Gernhard, 2008*); (b) assuming a relaxed clock and speciation under a Yule process (hereafter Relaxed + Yule), and (c) under a coalescent model of speciation assuming a constant population size (*Kingman, 1982*; hereafter Coalescent). All BEAST searches were carried out for $10 \times 10^6$ generations sampled every 1,000th generation using the GTR + Γ model of nucleotide substitution. Searches were evaluated for convergence using the criteria previously described for Bayesian phylogenetic reconstructions. Trees were summarized using TreeAnnotator v1.8.2 (https://beast.community/treeannotator) with 10% of trees discarded as burn-in and edges set using the mean-age option. Resulting ultrametric trees were analyzed in R using GMYC approach implemented by the 'splits' package (http://r-forge.r-project.org/projects/splits/).

## Morphological comparisons

A subset of specimens were observed and drawn using a Wild binocular microscope fitted with a 1.25X camera lucida with photographs taken using a Nikon D3100 DSLR camera fitted with a 95 mm macro lens. Individuals were sexed based on the presence of conspicuous dorsal horns in males and of a marsupium in adult females. Body lengths were measured from the anterior of the cephalon to the posterior tip of the pleotelson.

## RESULTS

Our final mitochondrial concatenated dataset consisted of 76 *Deto* individuals from 16 localities across the coastlines of both Namibia and South Africa (*COI* = 71 individuals, *16S rDNA* = 55 individuals, *12S rDNA* = 60 individuals). The final alignment of mitochondrial genes was 1,593-bp long after the exclusion of 26 nucleotide positions: 15-bp from the

*16S rDNA* and 11-bp from the *12S rDNA* alignments. The final *28S rDNA* alignment was 589-bp long after the exclusion of four nucleotide positions and included 30 individuals. All sequences produced in this study have been deposited in GenBank under accession numbers OQ852632–OQ852680, OQ870135–OQ870160, and OQ870100–OQ870134 (Table 1) and in the Barcode of Life Database (BOLD; BOLD BINs ADE2360 and ADV7260; specimens DECAC001-23 to DECAC076-23). Alignments with poorly aligned positions are provided as Dataset S1 and S2.

## Phylogenetic reconstructions

Preliminary phylogenetic reconstructions based on *COI* sequences including *D. marina* identified a moderately-supported basal split between two large *D. echinata* clades: a west coast "Namibia–Cape Town" clade (yellows and reds in all figures; Bootstrap support: 99 in Fig. 2), and a south coast "False Bay–Mossel Bay" clade (blues in all figures; Bootstrap support: 100). The "Namibia–Cape Town" clade included all individuals from localities in Namibia (A1–3), all those in South African samples collected between Jacobsbaai to Bakoven (A4–8), and one specimen collected from Simon's Town in False Bay (B2). The "False Bay–Mossel Bay" clade included all specimens collected in Kommetjie, South Africa (B1) the remaining specimens collected from Simon's Town in False Bay (B2) and all specimens collected to the east of this along the South African south coast (B3–B8).

Give the high level of divergence between *D. marina* and the in-group (Table 2), and the lack of sequence data for *D. marina* beyond *COI* sequences, we conducted our phylogenetic analyses on the concatenated mitochondrial dataset without *D. marina*, instead assuming a mid-point rooting scheme. All analyses identified the clades described above, however, the reciprocal monophyly of each clade was highly supported in all analyses (BS = 100 in all ML searches, MPP = 100 in all BI searches, Fig. 2). These analyses produced slightly improved resolution within the "Namibia–Cape Town" clade where we recovered the monophyly of all specimens collected in Namibian localities (A1–3) and one specimen collected in Simon's Town in South Africa (B2) (BS = 76–85; MPP = 81–99). Relationships within the "False Bay–Mossel Bay" clade remained poorly resolved.

*COI* K2P divergences amongst the two major *D. echinata* clades in our analyses ranged from 6.4–9.1% (Table 2); however, within-clade divergences were generally low with *COI* K2P divergences in the "Namibia–Cape Town" clade exhibiting divergences from 0.0–3.1% and those in the "False Bay–Mossel Bay" clade between 0.0–2.8%. For the former, the higher divergences were observed when comparing amongst Namibia and South Africa samples (*COI* K2P: 1.9–3.1%; Table 2).

## Haplotype networks

We identified 49 *COI* haplotypes separated by 88 segregating sites and 72 parsimony informative sites. Haplotypes segregated into two large networks separated by 39 or more mutational steps (Fig. 3). Network A contained all haplotypes recovered from specimens collected in localities in Namibia (A1–3), localities west of the Cape Peninsula (A04–A07), as well as one haplotype recovered from a single individual collected on the eastern coast of the Cape Peninsula, in Simon's Town (B02). Network B consisted of the remaining

Santamaria and Griffiths (2023), *PeerJ*, DOI 10.7717/peerj.16529
**Table 2 Estimates of evolutionary divergence between localities as indicated by COI K2P distances.** Minimum and maximum pairwise distances are provided with average distances in parenthesis. The number of base substitutions per site from between sequences are shown. Analyses were conducted using the Kimura 2-parameter model. This analysis involved 71 nucleotide sequences. There were a total of 659 positions in the final dataset. Evolutionary analyses were conducted in MEGA11.

| | A01 | A02 | A03 | A04 | A05 | A06 | A07 | A08 | B01 | B02 | B03 | B04 | B05 | B06 | B07 | B08 |
|---|---|---|---|---|---|---|---|---|---|---|---|---|---|---|---|---|
| A01 | 0.2–0.3 (0.2) | | | | | | | | | | | | | | | |
| A02 | 0.9–1.1 (1.0) | 0.0–0.0 (0.0) | | | | | | | | | | | | | | |
| A03 | 0.5–0.9 (0.7) | 0.8–1.1 (0.9) | 0.3–0.3 (0.3) | | | | | | | | | | | | | |
| A04 | 2.5–2.8 (2.7) | 2.8–3.0 (2.9) | 2.3–2.5 (2.4) | 0.0–0.3 (0.2) | | | | | | | | | | | | |
| A05 | 2.0–3.0 (2.6) | 2.3–3.1 (2.8) | 1.9–3.0 (2.5) | 0.3–2.6 (1.5) | 1.4–2.5 (1.8) | | | | | | | | | | | |
| A06 | 2.3–3.1 (2.8) | 2.6–3.1 (3.0) | 2.2–3.1 (2.6) | 0.2–2.8 (1.4) | 0.0–2.6 (1.6) | 0.5–3.0 (2.1) | | | | | | | | | | |
| A07 | 2.0–3.0 (2.5) | 2.3–2.8 (2.6) | 1.9–2.6 (2.3) | 0.2–1.5 (0.6) | 0.3–2.8 (1.6) | 0.3–2.8 (1.6) | 0.0–1.7 (0.9) | | | | | | | | | |
| A08 | 2.0–3.0 (2.7) | 2.3–3.1 (2.9) | 1.9–2.6 (2.4) | 0.2–1.5 (0.6) | 0.2–2.5 (1.4) | 0.3–3.0 (1.6) | 0.3–1.7 (0.9) | 0.2–1.7 (0.9) | | | | | | | | |
| B01 | 7.5–8.0 (7.8) | 8.0–8.4 (8.2) | 8.0–8.4 (8.2) | 6.8–7.3 (7.0) | 6.6–8.2 (7.5) | 6.9–8.3 (7.6) | 6.8–8.0 (7.3) | 6.6–8.4 (7.2) | 0.0–0.9 (0.5) | | | | | | | |
| B02 | 7.1–8.0 (7.5) | 7.6–8.4 (8.0) | 6.6–8.4 (7.6) | 6.4–7.3 (6.8) | 6.4–8.5 (7.4) | 6.6–8.5 (7.5) | 6.4–8.0 (7.0) | 6.4–8.4 (7.1) | 0.5–1.5 (0.9) | 0.0–1.2 (0.7) | | | | | | |
| B03 | 7.1–7.6 (7.5) | 7.3–8.0 (7.8) | 7.6–8.0 (7.9) | 6.8–7.3 (7.1) | 6.4–8.2 (7.5) | 6.9–8.2 (7.6) | 6.8–7.6 (7.2) | 6.4–8.0 (7.3) | 0.6–2.2 (1.1) | 0.3–2.2 (1.0) | 0.0–2.2 (1.1) | | | | | |
| B04 | 7.6–8.2 (7.9) | 8.2–8.5 (8.3) | 7.6–8.2 (7.9) | 6.9–7.5 (7.2) | 6.9–8.7 (7.9) | 7.1–8.7 (7.8) | 6.9–8.2 (7.4) | 6.9–8.5 (7.5) | 0.8–1.4 (1.0) | 0.6–1.4 (0.9) | 0.8–2.6 (1.3) | 0.0–0.3 (0.2) | | | | |
| B05 | 7.5–8.0 (7.9) | 7.8–8.4 (8.2) | 8.2–8.4 (8.3) | 6.6–7.3 (7.1) | 6.6–8.5 (7.8) | 6.8–8.5 (7.8) | 6.6–8.0 (7.3) | 6.6–8.4 (7.4) | 0.5–1.7 (0.9) | 0.0–1.7 (0.8) | 0.3–2.2 (1.1) | 0.8–1.5 (1.0) | 0.0–1.7 (1.0) | | | |
| B06 | 7.1–8.0 (7.7) | 7.6–8.4 (8.1) | 7.6–8.0 (7.9) | 6.4–7.3 (7.0) | 6.4–8.5 (7.5) | 6.6–8.5 (7.6) | 6.4–8.0 (7.2) | 6.4–8.4 (7.2) | 0.2–1.2 (0.9) | 0.0–1.5 (0.9) | 0.8–2.5 (1.2) | 0.0–1.4 (0.6) | 0.6–1.7 (1.0) | 0.2–1.2 (0.8) | | |
| B07 | 7.5–8.7 (7.9) | 8.0–9.1 (8.4) | 7.8–9.1 (8.3) | 6.8–7.8 (7.2) | 6.8–8.5 (7.7) | 6.9–8.5 (7.8) | 6.8–8.7 (7.4) | 6.9–8.7 (7.5) | 0.5–1.5 (0.7) | 0.0–2.2 (0.9) | 0.3–2.8 (1.1) | 0.5–2.0 (1.0) | 0.0–2.3 (0.8) | 0.6–1.9 (1.0) | 0.2–1.5 (0.8) | |
| B08 | 7.5–8.0 (7.8) | 8.0–8.4 (8.2) | 7.3–8.0 (7.7) | 6.8–7.3 (7.1) | 6.8–8.4 (7.8) | 7.0–8.4 (7.6) | 6.8–8.0 (7.3) | 6.8–8.4 (7.4) | 1.4–1.9 (1.6) | 1.2–2.2 (1.6) | 1.4–2.5 (1.7) | 0.9–1.4 (1.2) | 1.2–1.9 (1.5) | 1.1–1.9 (1.5) | 0.9–2.2 (1.4) | 0.0–0.3 (0.1) |
| *D. marina* | 16.0-16.2 (16.0) | 16.5–16.5 (16.5) | 16.2–16.2 (16.2) | 16.9–17.1 (17.0) | 16.7–16.9 (16.8) | 16.5–16.9 (17.3) | 16.9–17.1 (17.0) | 16.7–17.1 (17.0) | 16.0–16.7 (17.0) | 16.4–16.7 (16.6) | 16.9–16.9 (16.9) | 16.2–16.5 (16.4) | 16.5–17.6 (16.9) | 16.0–16.7 (16.5) | 15.6–16.7 (16.4) | 16.6–16.6 (16.6) |

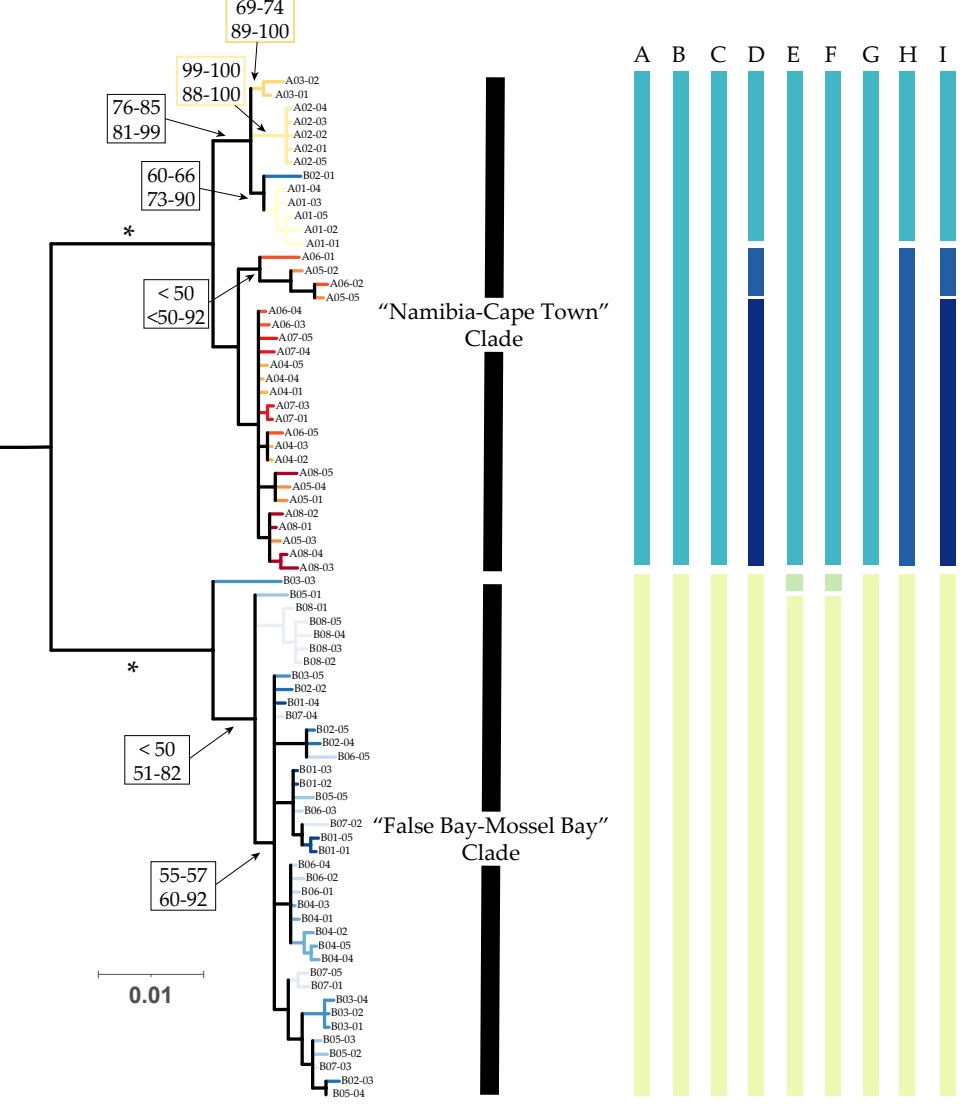

**Figure 2 Phylogenetic patterns of Deto from southern Africa.** Results are projected on the majority rule consensus tree produced by analyzing the concatenated mitochondrial dataset of *D. echinata* in RAxML partitioned by gene under the GTR + Γ model. We observed two monophyletic groups: a ''Namibia–Cape Town'' clade (blues and reds) comprised primarily of individuals collected from Namibia to Bakoven in South Africa, and a ''False Bay–Mossel Bay'' comprising individuals collected from Kommetjie to Glentana in South Africa. Values above branches represent support values for the corresponding branch (top value: Bootstrap Support; bottom: Maximum Posterior Probablities; *: 100 in all analyses). Results of molecular species delimitations are shown as bars, with members of the same putative species clusters identified by color. Bars correspond to: (A) ASAP analyses on the COI dataset, (B) ASAP analyses on concatenated mitochondrial dataset, (C) PTP on tree produced in RAxML assuming a single partition, (D) PTP on tree produced in RAxML assuming partitioning by gene, (E) PTP on tree produced in MrBayes assuming a single partition, (F) PTP on tree produced in MrBayes assuming partitioning by gene, (G) GMYC Relaxed+Yule, (H) GMYC Coalescent, (H) GMYC Strict+Yule.

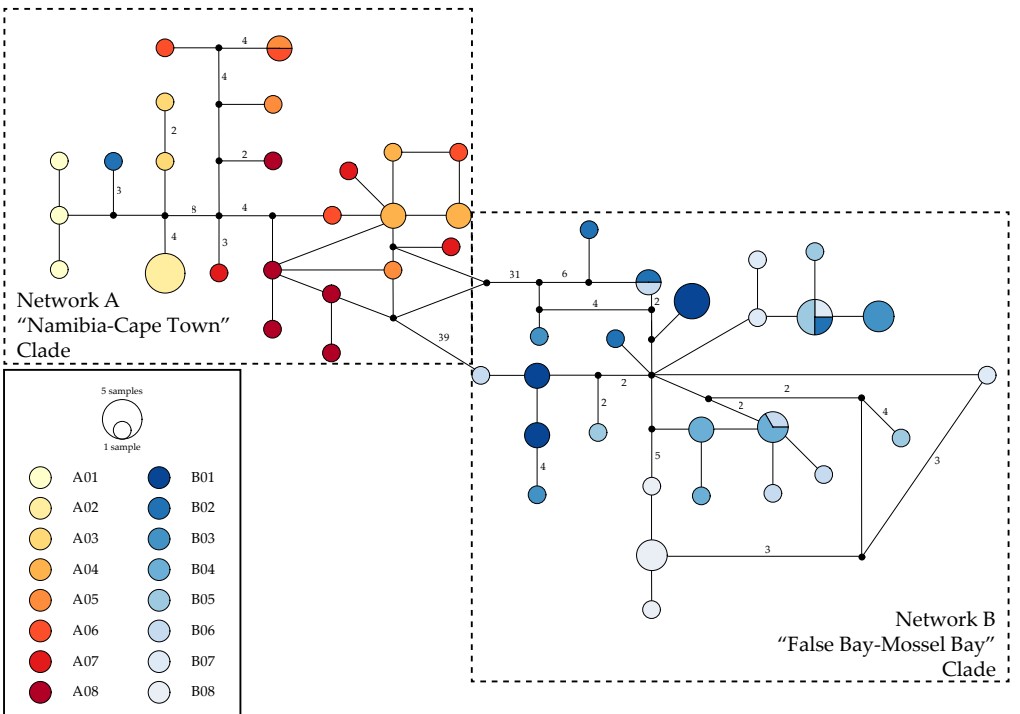

**Figure 3 Haplotype networks for the COI mitochondrial gene fragment of *Deto* from southern Africa.**
Colors correspond with those used in other figures. Black circles represent inferred unsampled haplotypes
with numbers along branches showing number of nucleotides differences between haplotypes. Frequency
of haplotype recovery is represented through the relative sizes of the circles. Locality labels correspond
with those in all other figures and tables. Network A corresponds to the "Namibia-Cape Town" Clade and
includes all individuals from localities A01-A08 as well as a sole individual from B02. Network B corre-
sponds to the "False Bay-Mossel Bay" Clade and includes all individuals from localities B01-B08 except
the B02 individual placed in Network A.

individuals collected from B02, as well as all other haplotypes recovered from individuals
sampled in localities from Kommetjie to Glentana, South Africa (B1, B3–B08). The two
networks were separated by >39 mutational steps, with separations within networks being
<20 steps.

We identified three *28S* rDNA haplotypes (Fig. 4): one shared by individuals collected
in localities in Namibia (A01–02), another shared by specimens from localities East of
the Cape Peninsula (B01–04, B07), and a third haplotype recovered from two individuals
from A05. The distance between these haplotypes was only two mutational steps, with the
haplotype found in A05 individuals equidistant to the other two haplotypes (Fig. 4).

## Molecular species delimitation analyses

Regardless of the dataset analyzed, ASAP analyses identified two putative species as the
most appropriate hypothesis for the number of species present among our *D. echinata*
specimens (Fig. 2, bars A and B). For the *COI* dataset, the two species hypothesis produced
an ASAP-score of 1.5, with a *p*-value of 0.381 (rank = 2), a W value of $1.25 \times 10^{-4}$ (rank
= 1), and a threshold distance of 0.041458. For the concatenated mitochondrial dataset,

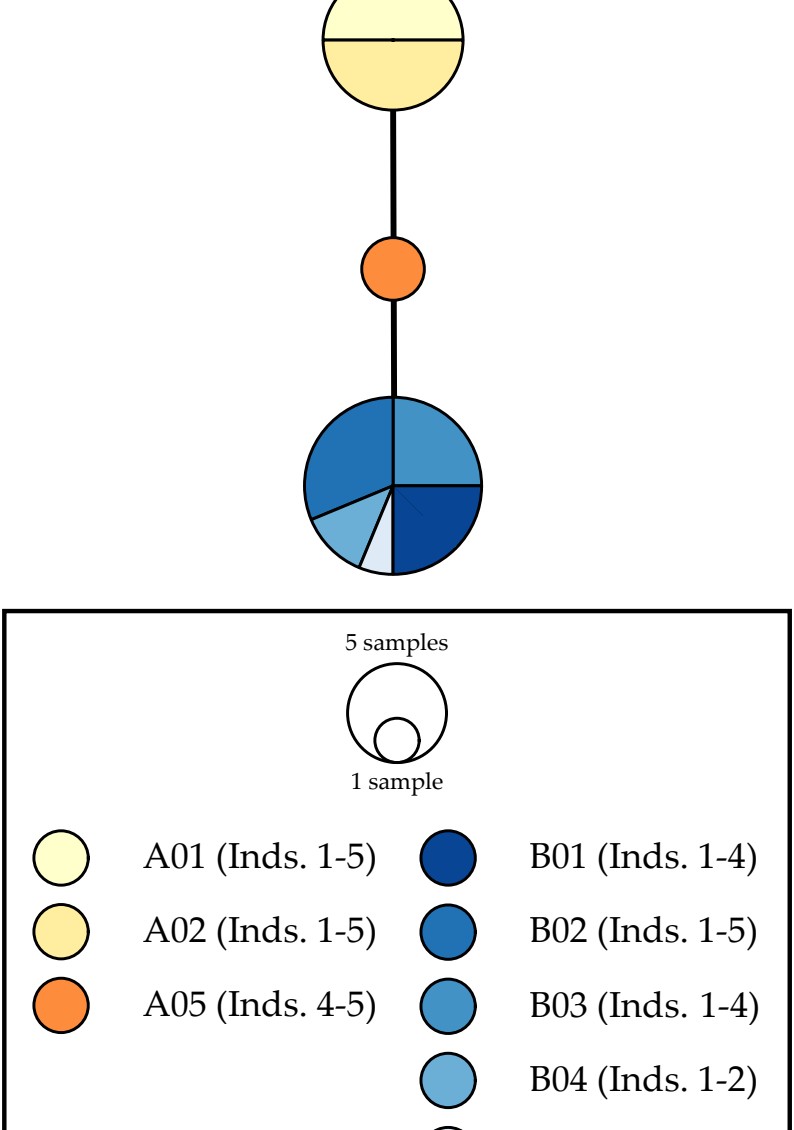

**Figure 4** **Haplotype networks for the nuclear 28S rDNA gene fragment.** Colors correspond with those used in other figures. Frequency of haplotype recovery is represented through the relative sizes of the circles. Locality labels correspond with those in all figures and tables.

ASAP identified the two species hypothesis as the highest ranked solution with an ASAP score of 2, a $p$-value of 0.244 (rank = 1), a W value of $1.88 \times 10^{-4}$ (rank = 3), and a threshold distance of 0.028554.

PTP and GMYC analyses identified two to four putative species clusters (Fig. 2). PTP analyses identified two putative species clusters when analyzing the phylogeny

produced in RAxML assuming a single partition (Fig. 2, bar C) and four clusters for the phylogeny produced when assuming partitioning by genes (Fig. 2, bar D). Meanwhile, PTP analyses identified three putative species clusters for the phylogenies produced in MrBayes, regardless of partitioning used. For GMYC analyses, the Relaxed + Yule analysis identified two putative species, the Coalescent analysis identified three putative species, and the Strict Clock + Yule analysis identified four.

Results of MSDAs were largely congruent both in the number of putative species clusters identified and the assignment of individuals to clusters with differences between analyses due to splitting of larger clusters found in more inclusive analyses. Both ASAP analyses, the PTP analysis based on the unpartitioned ML phylogenetic reconstruction, and the GMYC Relaxed Clock + Yule analysis identified two putative species clusters. Each of these clusters corresponded to one of the major clades recovered in our phylogenetic reconstructions, with all "Namibia–Cape Town" clade individuals grouped in a single putative species and all "False Bay–Mossel Bay" clade individuals placed in a separate putative species (Fig. 2). Both PTP analyses of MrBayes phylogenetic reconstructions recovered a species cluster consisting of all "Namibia–Cape Town" individuals and two species consisting of "False Bay–Mossel Bay" individuals, one consisting of a single individual from Kalk Bay (B03) and the other containing the rest of the members of the clade. The remaining analyses identified a single species that consisted of all "False Bay–Mossel Bay" individuals, differing from the two species cluster solution by the splitting of "Namibia–Cape Town" individuals into two to three species depending on the analysis (Fig. 2, bars D, H, and I).

### Morphological comparisons

Although we made initial observations on the morphological appearance of specimens from the two recognized clades, we were unable to detect any consistent features which could be used to separate the two 'species'.

## DISCUSSION

### Cryptic diversity

Molecular approaches have led to the discovery of cryptic genetic diversity in various coastal invertebrate taxa from southern Africa, including gastropods (*Evans et al., 2004*; *Zardi et al., 2007*), the annelid *Arenicola loveni* (*Simon et al., 2020*), the barnacle *Tetraclita serrata* (*Reynolds, Matthee & Heyden, 2014*), the amphipod *Talorchestia* (now *Africorchestia*) *capensis* (*Baldanzi et al., 2016*), and various isopod genera, including *Tylos* (*Mbongwa et al., 2019*; *Bezuidenhout et al., 2021*), *Excirolana* (*von der Heyden, Mbongwa & Hui, 2020*), *Exosphaeroma* (*Teske et al., 2006*), and *Ligia* (*Greenan, Griffiths & Santamaria, 2018*). For the latter, molecular studies have reported the lineages within species exhibiting divergences similar to those we report here for *D. echinata*: 3.1–12.0% *COI* K2P for lineages in *Ligia glabrata* and *Ligia natalensis* (*Greenan, Griffiths & Santamaria, 2018*), up to 10% *COI* K2P in *Tylos granulatus* (*Mbongwa et al., 2019*), and 17–18% uncorrected-p for two *Excirolana* species (*von der Heyden, Mbongwa & Hui, 2020*). Thus, our discovery of two highly divergent genetic lineages within *D. echinata* represents another instance of cryptic diversity in invertebrates from the region.

The two *D. echinata* lineages reported herein exhibit amongst lineage *COI* K2P divergences that exceed 6.0% (Table 2). These divergences greatly exceed the broad cut-off suggested to delineate between within and amongst species genetic distances for metazoa (*Hebert et al., 2003*) and crustacean species (*Matzenda Silva et al., 2011*; *Raupach et al., 2015*). Indeed, the genetic distances amongst the lineages reported match, or exceed, those reported for other valid species of coastal isopods (*e.g.*, *Hurtado, Lee & Mateos, 2013*; *Santamaria et al., 2013*; *Santamaria et al., 2017*; *Hurtado et al., 2018*), including recently described cryptic species (*e.g.*, *Santamaria, 2019*). The two lineages also harbored different and unique 28S rDNA haplotypes, suggesting that they also exhibit differences in nuclear loci. Thus, our findings suggest *D. echinata* is a cryptic species complex in need of taxonomic revision.

Our exploratory MSDAs suggest that *D. echinata* represents at least two species, as most MSDAs identified the members of each highly divergent genetic lineage recovered in phylogenetic reconstructions as corresponding to separate species and BOLD recognized two separate BINS in our dataset. Thus, we suggest *Deto* populations from Namibia to Bakoven, South Africa are likely to represent one species, with populations from Kommetjie to Glentana, South Africa representing the other.

Despite convincing genetic evidence of *D. echinata* being a cryptic species complex, we did not observe diagnostic differences between these putative species in three morphological traits used in *Deto* taxonomy such as horn length, horn shape, and positioning of pleopods. However, horn length and shape are known to differ not only between male and females, but also between individuals of different sizes, and between individuals differing in body condition (*Glazier, Clusella-Trullas & Terblanche, 2016*). Thus, the absence of differences between the two clades in our preliminary morphological evaluations may be due to the confounding effects of sex, size, and/or body condition among the collected samples. These confounding effects may also account for previous taxonomic confusion in *D. echinata,* such as the erection and subsequent synonymization of *D. acinosa* based on differences in horn size. Given the complexity of this situation, we thus have elected not to present our inconclusive preliminary morphological observations here. Rather, we suggest that a separate comprehensive taxonomic work is needed to determine whether diagnostic morphological differences exist between the *D. echinata* lineages herein reported.

We suggest that future taxonomic revisions consider the two clades recovered in our phylogenetic analyses as hypothetical or putative species. Future taxonomic work should also incorporate outlying *D. echinata* populations, as these isopods are reputed to occur in Saint Paul Island in the Southern Indian Ocean (33°S, 77°E), midway between South Africa and Australia. Given its distance to the localities included in this study, it is likely that this outlying population represents another unique genetic lineage, different from those reported herein.

## Phylogeographic patterns

Molecular studies on coastal invertebrates in South Africa have uncovered complex phylogeographic patterns not congruent across taxa, with related species exhibiting differing patterns (*e.g.*, *Teske et al., 2006*; *Reynolds, Matthee & Heyden, 2014*; *Mmonwa*

*et al., 2015*; *Baldanzi et al., 2016*; *Greenan, Griffiths & Santamaria, 2018*; *von der Heyden, Mbongwa & Hui, 2020*). Despite contrasting patterns, most of these poorly dispersing coastal invertebrates exhibit strong phylogeographic breaks around the Cape of Good Hope, which is regarded as the most significant biogeographic transition zone around the entire South African coastline (*Griffiths et al., 2010*). *Greenan, Griffiths & Santamaria (2018)* saw a break in the distribution of *Ligia* lineages, with a major lineage distributed from Ganzekraal, South Africa (A05 herein) to Luderitz, Namibia (A03) and another lineage being distributed from Kommetjie (B01) to L'Agulhas in South Africa. *von der Heyden, Mbongwa & Hui (2020)* also found a break in the distribution of lineages for two *Excirolana* species between localities in the west of Cape Point and those in False Bay and eastward, with the phylogeographic break occurring in the region between Yzerfontein and Simon's Town (B02) in False Bay. Meanwhile, *Teske et al. (2006)* identified two major and non-overlapping lineages within the sphaeromatid isopod *Exosphaeroma hylecoetes* from the regions around the Cape Peninsula: a south-western lineage that included individuals collected in estuarine localities between Cape Agulhas and the Cape of Good Hope, and a western lineage from localities west of the Cape of Good Hope. The two highly divergent genetic *D. echinata* lineages uncovered in this study are no exception, as the two lineages we uncover exhibit a strong phylogeographic break in the region, with the "Namibia–Cape Town" and the "False Bay–Mossel Bay" exhibiting a break in distribution somewhere in the area Bakoven (A07) and Kommetjie (B01).

Despite the strong phylogeographic break we report herein and the poor dispersal capabilities of *D. echinata*, we did observe evidence suggesting that rare dispersal events may take place across the Cape of Good Hope. A single *D. echinata* specimen collected in Simon's Town (B02), a location in False Bay east of the phylogeographic break, harbored mitochondrial haplotypes that placed it in the "Namibia–Cape Town" clade. Interestingly, this individual shared *28S* rDNA haplotypes with individuals from the "False Bay–Mossel Bay" clade. These patterns suggest this individual from Simon's Town may represent a hybrid, or a recent migrant harboring an ancestral polymorphism at the *28S* rDNA gene.

Phylogeographic patterns of *D. echinata* exhibit some important differences to those exhibited by other coastal isopods in the region. For instance, *D. echinata* populations do not appear to exhibit the high levels of population differentiation and structuring reported for *Ligia* isopods (*Greenan, Griffiths & Santamaria, 2018*) and *Tylos granulatus* (*Mbongwa et al., 2019*; *Bezuidenhout et al., 2021*). Whereas genetic divergences amongst populations within lineages in *Ligia* from southern Africa ranged exceed 5.0% *COI* K2P in many instances (*Greenan, Griffiths & Santamaria, 2018*), *COI* K2P divergences amongst localities of *D. echinata* within the same lineage rarely exceed 2.5%. The exception to this pattern being the relatively higher levels of differentiation seen amongst Namibia populations and South Africa populations within the "Namibia–Cape Town" lineage (1.9–3.1% *COI* K2P) indicating at least a potential effect of distance in the isolation and genetic differentiation of *Deto* populations in southern Africa.

The presence of phylogeographic breaks in the Cape Peninsula region for these taxa, appear to suggest that the area's geological history may be the primary driver of the distribution of genetic diversity for coastal organisms in the area (*Toms et al., 2014*).

Subtle differences in the location of phylogeographic breaks in this region; however, may reflect differences in life history, evolutionary history, and ecological needs amongst taxa (*Pelc, Warner & Gaines, 2009*). The Cape Peninsula region is known to exhibit ecological gradients in both biotic and abiotic factors, as a result of the varying influences of the colder Atlantic waters and warmer Indian Ocean to the east and west of the Cape Peninsula respectively (*Bustamante et al., 1995*; *Demarcq, Barlow & Shillington, 2003*). Indeed, *Baldanzi et al. (2016)* suggested that the unusual phylogeographic break seen in the amphipod *Talorchestia* (now *Africorchestia*) *capensis* in the Cape region may be due to species-specific environmental conditions and requirements. Differences between the biology of organisms may thus help explain the location of phylogeographic breaks reported for different organisms in the Cape region. For instance, the distribution of lineages of air-breathing organisms that rarely submerge, such as *Deto*, *Ligia*, *Tylos*, and *Talorchestia* may be more likely influenced by terrestrial climatic conditions, while the distribution of underwater breathing organisms, such as *Exosphaeroma,* are more likely influenced by marine environmental conditions. Relatedly, differences in climatic conditions as well as climate oscillations may explain the differences in divergence and divergence time estimates reported for coastal organisms in the Cape region (*Teske et al., 2007*; *Teske et al., 2009*). In the case of *Deto*, the moderate level of divergence between the two major clades uncovered suggest their origin likely predate the Last Glacial Maxima. Rough estimates of the divergence time for the two major lineages of *D. echinata* reported herein using the ∼6.4–9.1% COI K2P divergences seen amongst lineages and COI substitution rates reported for a variety of isopods (0.0125 substitutions per site per million years reported by (*Ketmaier, Argano & Caccone, 2003*) from *Stenasellus* isopods; 1.56–1.72% divergence per million years reported by *Poulakakis & Sfenthourakis (2008)* from *Orthometopon* isopods) suggest *D. echinata* lineages diverged at least 2.4 million years ago. This suggests that Plio-Pleistocene climactic and oceanographic events (*e.g.*, sea level changes) may have shaped the evolutionary history of *D. echinata* in the region. Nonetheless, given the poor state of ecological knowledge of South African beach (https://www.sciencedirect.com/topics/agricultural-and-biological-sciences/macrofauna), concerted efforts should be made to better account for the effect of ecological variation in shaping population dynamics of marine species in the region.

## ACKNOWLEDGEMENTS

We would like to express our gratitude to Dr Maya Pfaff of the South African Department of Environmental Affairs, Forestry and Fisheries for collecting specimens from the Namaqualand National Park and to members of the Santamaria laboratory for help in conducting molecular work.

### Funding

Financial support for work done by the Griffiths laboratory was provided by a research grant from the University of Cape Town Research Committee, while work completed in

the Santamaria laboratory was supported by start-up funds at the University of Tampa and the University of South Florida Sarasota-Manatee. The authors received no external funding for this work. The funders had no role in study design, data collection and analysis, decision to publish, or preparation of the manuscript.

### Grant Disclosures

The following grant information was disclosed by the authors:
University of Cape Town Research Committee.
University of Tampa.
University of South Florida Sarasota-Manatee.

### Competing Interests

Carlos A. Santamaria is an Academic Editor for PeerJ.

### Author Contributions

- Carlos A Santamaria conceived and designed the experiments, performed the experiments, analyzed the data, prepared figures and/or tables, authored or reviewed drafts of the article, and approved the final draft.
- Charles L Griffiths conceived and designed the experiments, performed the experiments, authored or reviewed drafts of the article, and approved the final draft.

### Field Study Permissions

The following information was supplied relating to field study approvals (i.e., approving body and any reference numbers):

Field collections were carried out under Scientific Collection Permit RES2017/53 issued by the South African Department of Environmental Affairs.

### DNA Deposition

The following information was supplied regarding the deposition of DNA sequences:

All sequences produced in this study are available at GenBank: OQ852632–OQ852680, OQ870135–OQ870160, and OQ870100–OQ870134.

### Data Availability

The concatenated alignments with poorly aligned positions removed are available in the Supplemental File.

### Supplemental Information

Supplemental information for this article can be found online at http://dx.doi.org/10.7717/peerj.16529#supplemental-information.

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
