# Peer review of "Cryptic diversity and phylogeographic patterns of Deto echinata (Isopoda: Detonidae) in southern Africa"

_PeerJ, doi:10.7717/peerj.16529_

## Round 0.1 · original submission · Major Revisions

Apart from the criticisms raised by the Reviewers, I have a major question, regarding the analysis of nuclear 28S rDNA. Being nuclear, it expected to be diploid and therefore to harbour instances of heterozygosity. So, how was haplotyping performed/inferred?

Besides, being an RNA (and not protein) coding gene, it seems to me to make no sense the use stop codons to filter pseudogenes.

I also share the worry about the use of a single nuclear genomic region (although I do not necessarily advise the use of ddRADseq)

I do hope the Authors can successfully address our criticisms.

·

Basic reporting

This is a well-prepared MS presenting a decent phylogeographic analysis of an isopod 'species'. All parts of the MS are carefully structured and written, with a good use of literature. The phylogeographic work is based on a set of 3 mtDNA and 1 nuclear DNA sequences, which are adequate for revealing the cryptic diversity suggested by authors, even if today it should be useful to use a more advanced 'genomics' approach (e.g. ddRADseq) that could provide a richer source of data, allowing also population-level insights. Nevertheless, the analysis is ok as is, given the scope of this work. I have nothing special to comment on (besides a secondary detail in line 262 where the acronym MSDA appears for the first time - it needs an explanation in Methods).

Experimental design

No comment

Validity of the findings

No comment

Additional comments

No comment

Reviewer 2 ·

Basic reporting

Well communicated paper, issues primarily with the data analyses and types of analyses performed.

Experimental design

1) How is the sampling justified. Between Saldanha/Langebaan on the west coast there are no other sampling localities in SA? Why is this? This is poor sampling design or indicative of “lets sample what we have”
2) Why did you use 28S as a nuclear marker? Its super conserved and does not reflect the mt/ nu discordance
3) On the tree topology (Fig 2) you need to show the two clades – label them as you did with the COI network and say where they are distributed
4) I would like the authors to also run PTP and GYMC on the COI data for the species delimitation – plot these results onto the tree topology and then DISCUSS the results, how useful are these species delimitation results
5) If you have two species, what is the morphological results for this? Do you observe morphological differences in the two clades?
6) I would like to see a divergence time estimation based on mutation rates on the tree topology so that a better understanding can be gained about when the divergence occurred and the potential abiotic features are that drove cladogenesis in the group.

Validity of the findings

Well interpreted within the context of the finding but needs additional analyses. Please see above.

·

Basic reporting

1. The article is well-written, and the dataset constitutes a substantial body of work.

2. Lines 76-81 (Introduction) are near-identical to lines 272-279 (Discussion). Please modify the text so there is not such exact overlap. (Typically, the introduction would provide more general background, and the discussion in more detail where relevant).

3. Lines 282-312 (Discussion) here the authors present at length the detail of previous studies. I suggest some of this information, which compares and contrasts the various proposed species boundaries, could also be presented as part of the existing map figure e.g. showing where the current study is drawing the 'phylogeographic break' vs where others have found similar/different phylogeographic breaks. I think this will make the discussion easier to follow. Currently, these sites that are specified from others' studies are rather out of context for the reader yet indeed it is building the argument for the findings of this paper.

Furthermore, minor comments to these two paragraphs:
- Line 283: could remove “even”, and specify what level of relatedness is being referred to here
- Line 287: “proposed biogeographic barriers” – who proposed, and which barriers exactly?


4. Figure 1: I find that the way the sites are subdivided for the inset (which is presumably to distinguish between the close-together sites, to allow space for the labels) compounds the issue that the reader faces -- to imagine where the various cryptic species boundaries are, for the current research and the previous boundaries that are discussed from other studies. It would be ideal if the sites A01 – A07 could be on the main map image, and sites B01 – B08 provided in the inset. To minimize this confusion. Furthermore, please indicate on this map image the division and clade labels for the two species groups that you propose in your findings. If possible, please also add the biogeographic boundaries that other groups have proposed – see my comment #3 above. Please also add latitude and longitude markings.

5. Figure 2: Given the division into two species-level groups, and the geographic separation of these, as the main findings of this paper, why is it that there is a bright green individual B02-01 within the "A" clade? This is not discussed? e.g. do you think it is a recent arrival to the "A" area? Please also add to this figure, the two clade labels that are mentioned in the caption

6. Figure 3: Consider adding the two distinct species regions as titles to the legend list for sites A and sites B

7. Line 325 (discussion): I suggest to remove “with their different biological and ecological needs and tolerances”, as this sounds rather weak. Would it be relevant here -- to instead comment on the poor dispersal abilities of these coastal isopods (What characteristics of these taxa make them poor dispersers? Compared to which other taxa that are better dispersers?), as this point about poor dispersal ability seems to be emphasized in the introduction but then not picked up on later. If not in this last paragraph, then perhaps elsewhere in the discussion. I find this concept rather interesting though, that the population genetic structure in the region is thought to be mostly resulting from geological histories (as they are poor dispersers?), yet the ecological differences between species may account for the differences in the genetic species boundaries. It’s a nice way to end the manuscript.

8. Figure 4: please remove the caption text about the black circles and line numbers representing the number of nucleotides between haplotypes. This has presumably just been copied from the previous caption and isn't relevant here because there is only a single nucleotide difference between each of the three haplotypes (no black circles and no line numbers).

Experimental design

1. The abstract states additional goals, which are not clearly outlined in the last paragraph of the introduction. The goal of assessing whether the phylogeographic distribution of D. echinata matches those of other coastal peracarids in the region is not really addressed at the analysis level. I suggest to include this information to the figures (e.g. as a hand-drawn layer on the map) to indeed show how the findings relate to sister taxa in terms of species boundaries and cryptic diversity in the region.

2. I had not previously heard of the ASAP method for species delineation (typically could otherwise use a combination of BOLD barcode index numbers, and generalised mixed yule coalescence (GMYC)). But I find the authors choice of methods overall to be appropriate, as there is a clear distinction between the two putative species groups.

3. Lines 102-104: Is there another section that these details of collection permit could be shifted to?

4. Line 143: please specify approximately the date that you performed the search for public data

5. Line 252: please add the % divergence (e.g. could be just for COI) explicitly here, to substantiate the statement of your findings.

6. Line 262: What is meant by "exploratory MSDA analyses" ? This is the first mention of the acronym.

Validity of the findings

Based on mitochondrial and nuclear gene regions (COI, 16S rDNA, 12S rDNA, 28S rDNA) the ASAP analysis consistently grouped the populations into two geographically-distinct species.

1. Lines 240-242: please include these clade labels (“Namibia-Cape Town” and “False Bay-Mossel Bay”) also on the figures throughout. (This may relate to some of my previous suggestions)

2. The sequences are not yet available on GenBank, although presumably the provided accession codes will become live after publication is accepted.

3. Please also deposit at least the COI sequences to the barcode of life datasystems (BOLD) online repository, along with any associated metadata (specimen images, GPS locations, etc). e.g. in the form of a publicly-available dataset for this paper.

Reviewer 4 ·

Basic reporting

The overall quality of the manuscript was good, but I have a lot of suggestions to improve it in a number of places, including the quality of the figures, which needs improving:

Abstract.
Line 17-20: I suggest combining the first sentence with the first part of the second sentence to make this statement more succinct, given they are saying the same thing really, but one is specific to South Africa or remove the first sentence entirely.
Line 26-27: it is not clear what is meant by “cryptic genetic diversity”
Line 28. “coastal peracarids” to “coastal invertebrates” would be better.
Line 30: …. findings suggest D. echinata “to be” to “is”

Introduction
Line 34: “work on” to “studies of”; “has” to “have”
Line 39: Mention the various different groups of invertebrates, at least to order level, that have been studied. Also some further information about the phylogeographic patterns and hypotheses about how phylogeographic structure was generated would be good to present in the introduction, particularly if it is likely to be relevant to the D. echinata study.
Line 51: give the authority of the genus and species when first mentioned.
Line 56: …….unknown and it has not been reported since it original and…” “it” to “its”
Line 84: reference to a map figure would be useful for explaining the distribution of D. echinata. Indeed Fig. 1 could be mentioned here, and adjusted to show the current known distribution of the species as it is not cited at all in the entire manuscript.

Line 91-92: as mentioned above, it is not clear what is meant by “cryptic genetic diversity”. “Cryptic” is a term often used to highlight species that show no morphological variation, but are divergent for genetic markers.
Line 93-4: why restrict the comparison to coastal isopods only if there are other studies of coastal animals that have been made? Indeed, I read later that the taxa compared are much broader. From these past studies are there any hypotheses for the phylogeographic patterns that could be proposed for testing using the D. echinata data? This doesn’t need to be detailed, but just outline that phylogeographic structure has been commonly found, with a break around the coast, associated with some aspect of the geological or climatic history of the region. Presumably sea levels also dropped in the past (during ice age glacial maxima), and the coastal position of the species may have been very different.

Methods
Line 100: refer to Fig. 1 here to highlight sampling locations. It is useful to show that the sampling covers the distributional range of the species as much as possible. Also, did the sampling include individuals with reduced horns to allow the assessment of the existence of D. acinosa? If so, it would be good to include this information somewhere here or in Table 1.
Line 111: generally gene symbols are usually italicised

Line 119 “pseudo-genes” to “pseudogenes”
Line 120: give the web site for obtaining Geneious.
Line 130: delete second “levels”
Line 157 -163: What model of nucleotide evolution was used in the analyses. How many MCMC generations were run in the Bayesian analyses?
Line 160: “sampled” to “sampling”
Line 215-223: It is not clear why a network analysis needs to be presented for COI, given the phylogenetic analysis presented in Fig. 2.

Results
Line 177: It would be good to know how many individuals were sequenced for the different markers used, including 28S. At the moment, only the final concatenated mtDNA data set is mentioned, but COI data were singled out for separate analyses too.
Line 189 and Line 202: Refer specifically to Fig. 2 in the text here as it is not mentioned anywhere else. Also, if the tree shown is an ML tree then it would be good to include the BI tree as a supplementary figure.

Discussion
Line 246: As per comment above, please clarify what “cryptic genetic diversity” is.
Line 255-257: I don’t think comparing the divergence to those of metazoans provides a good comparison. Crustaceans, particularly isopods, often have deep genetic divergence within species, so it would be better to focus on this group if proposing that a genetic distance of a particular amount equates to a distinct species. See paper by Lefubere and colleagues (2006, MPE) for further information about crustacean divergences
Line 262: what is MSDA? It doesn’t appear to have been defined anywhere. If this is the ASAP method that was applied, it should be noted that it will not distinguish phylogeographic structure within species from the presence of two species. With respect to the evolutionary independence of the two genetic lineages, it would be worth discussing what was found with the 28S data too. Also, what does the finding of a haplotype B in the A clade say, if anything, about the presence of different species. Does this represent a case of introgression or dispersal of this B species into the range of the A species? MtDNA data, doesn’t really help here, but if it has a 28S sequence as well then that would be interesting to know, particularly if it is a B haplotype in an individual with an A mtDNA type.

Line 299-312: This discussion is interesting, but the mention of so many place names without reference to a map, makes it very difficult to follow for a non-South African. Perhaps focus on the concordant patterns with that found for D. echinata, rather than giving all the known patterns for taxa with coastal distributions and refer to Fig. 1 for a map showing some of the localities. It would also be great if there was more focus on the factors that likely drove the phylogeographic patterns.

Line 318: “ranged exceed” is unclear.
Lines 325-7: What is the geological history that is likely to be relevant here? How would geological history impact coastal species? This raises the question why some molecular clock analyses weren’t conducted as part of this study, to obtain an estimated date of divergence of the two mtDNA lineages. There are biogeographic calibrations available for isopods, which could be used to obtain a rough date of coalescence for the two mtDNA lineages and this could then be compared to the proposed geological history or some aspect of the climatic history of the region.

Figure 1: placing some of the key place names on the map would be useful for the reader (e.g. A01, A02, A03 and B08), especially to highlight where the key phylogeographic break is located too.

Figure 2: Modify the title to be clear what is presented in this figure. Was it the ML tree or the BI tree that was shown? Was it based on concatenated mtDNA data from COI, 12S and 16S? Mention that the data come from D. echinata, rather than Deto generally. Posterior probabilities are usually given as a fraction out of 1, given that it is a probability, not a percentage. This should be modified in the figure too. Showing the map inset here from Figure 1 with the position of the phylogeographic break would also improve the quality of the figure, and help the reader work out where the population codes are located.

Figure 3: it is not clear what this haplotype network adds to the paper, that the phylogenetic figure 2 already provides. I suggest this should be moved to supplementary information if it is mentioned at all in the text.

Figure 4: The frequency of the 28S individuals in each circle should be provided, as it is impossible to infer this from the small circle legends that were provided. With the removal of Fig. 3, this figure could also be combined with the phylogenetic figure (Fig. 2), given that it doesn’t need to be a large figure to show 3 circles.

Table 1: why are there so many N/A for the latitude and longitude of the different sites? Surely this information could be made available, or provide rough lats and longs if precise values were not obtained at the time.

Table 2: It is not clear in the title what these divergence values are based on. If it is based on COI sequence data, it is not clear where “non coding” positions come from, as the Folmer primers PCR-amplify the protein coding region of COI, and any positions upstream would be part of a tRNA, so they are coding for something.

Experimental design

This manuscript by Santamaria and Griffiths, presents a phylogeographic analysis of the coastal isopod Deto echinata in southern Africa. Their analyses of mitochondrial data from 3 genes and a nuclear marker provide evidence for a phylogeographic break in this species, potentially indicative of cryptic species being present. The overall findings of phylogeographic structure are solid, despite a relatively small data set being presented, with respect to the number of individuals being analysed.

The authors compare their results with analyses of other coastal invertebrates in southern Africa, but do not go into much detail about what has potentially driven the phylogeographic structure of D. echinata. I think there is great potential in this paper to go further with the analyses and present an approximate time when the two major lineages diverged using molecular clock methods (e.g. using BEAST), calibrated with an isopod COI divergence rate (available from biogeographic studies). This would be an improvement of some of the past studies of phylogeographic patterns only and allow some of the biogeographic hypotheses to be potentially explored.

Validity of the findings

As mentioned above the evidence for phylogeographic structure is solid, but the factors that might have led to the phylogeographic structure are not really explored, because of the lack of any molecular dating analysis. This means the study mostly presents a phylogeographic pattern and a suggestion that there are cryptic species present. For the latter it should be suggested that analyses of morphology too should be made, and finer scale analyses of possible introgression at the boundary should be made to verify the hypothesis that the taxa are morphologically cryptic and distinct species.

Additional comments

no comment

---

## Round 0.2 · Minor Revisions

I thank the Authors for the improvements on the previous version.
However, I share the criticisms raised by Reviewer #4 and I could not find the rebuttal of my previous comments.

Please provide a detailed, point-by-point, answer to the raised questions.

Reviewer 2 ·

Basic reporting

Fine

Experimental design

Fine

Validity of the findings

Solid and scientifically sound

Additional comments

None

·

Basic reporting

Edits to the manuscript in response to reviewer comments are great. The authors addressed all of my comments with thought and consideration, and the writing style/quality is excellent. No further edits to suggest from my side.

Experimental design

The additional species delimitation methods add validity to the results and the presentation of these results in Figure two is nice

Validity of the findings

no comment

Reviewer 4 ·

Basic reporting

This is my second review of the manuscript (reviewer 4 previously). Overall, I think the authors have significantly improved their paper, but there are still a few parts of the paper where the writing could be improved for clarity or where further information could be provided. It was also a little frustrating that the authors often ignored some of my suggestions in their rebuttal, while claiming in the rebuttal that they had addressed the issue. I recommend that they at least comment on each suggestion, even if they dont want to make a change to their manuscript.

I still think the first part of the abstract could be better worded. The change I suggested wasn’t actually incorporated, which is fine. However, parts of the two sentences are unclear and it would be good to say what it is specifically meant by “complex patterns” and “contrasting phylogeographic patterns”. Is it that there are multiple phylogeographic lineages within species, which are not coincident in time or space? Alternatively, leave these parts out of the sentences (they can be explained later in the introduction) and focus on the “cryptic diversity”.

Similarly, my suggestion of changing “work” to “studies of” (line 44 of the track changed document) was ignored too. “works” is an odd word to use here.

Line39 suggestion – this suggestion was ignored in the rebuttal, but statements like “contrasting phylogeographic patterns” don’t specifically say a lot. As asked above are there multiple phylogeographic lineages within species and what is known about where the barriers are around this coastal section. Are they coincident? I have seen that barriers are now shown in Fig. 1, which is great, so why not actually provide some background information about them in the introduction?

Comment to Line 93-4: again this comment was largely ignored, and little information was actually provided about the other isopods in the introduction. For example, how many distinct phylogeographic lineages were found in other Ligia spp. and where were they distributed? I realise that there was more detailed information provided in the discussion, but it is a good practice to present some of the biogeographic hypotheses in the introduction too, such as where some of the major phylogeographic breaks occur from past studies of species with similar dispersal capabilities and ecological requirements.

It was good to see some additional species delimitation analyses. For the new sections I suggest:
Line 350 and 354 (track changed document): add the word “analyses” after “PTP”.
Line 356: “In GMYC” to “For GMYC analyses”

From line 423 (track changed document) the authors could also mention there is some support from the nuclear marker 28S too for the two cryptic species, because they had different 28S haplotypes. This suggests it is not just a single genetic locus (ie. mtDNA) that supports the distinction of two cryptic species.

Dating analyses: the authors have chosen not to present their molecular clock analyses, and I can partly understand the reluctance, given the error margins around the estimates potentially being very high. However, trying to place a time on divergences, rather than just stating a COI divergence level, is much more meaningful, even though the confidence intervals around the estimates are broad. At a minimum the authors should at least do a basic calculation, even if they don’t want to do a molecular clock analysis (though, note that they have already done the hard work with some BEAST analyses).

For example, they could simply say that the minimum 6% divergence level among haplotypes from the two major lineages of D. echinata would equate to approximately 2.4 million years of separation, using a borrowed rate of evolution for COI of 0.0125 substitutions per site per million years for subterranean aquatic stenasellid isopods (Ketmaier, Argano, & Caccone, 2003). If they have other isopod calibrations, they could quote these different divergence estimates too. This then becomes a point of discussion because it suggests an influence of Plio-Pleistocene climatic changes that may have periodically isolated populations of the isopod around the coastal regions of southern Africa. By just quoting a COI divergence level there are only distributional patterns left to discuss, and it is unclear if they are possibly coincident in time or not with what is found in other species.

Experimental design

I have no further comments to add. It was good to see some different species delimitation methods used, but I would prefer to see them present the dating analysis to provide some kind of estimate for the coalescence time tof the phylogeographic lineages.

Validity of the findings

no comment

Additional comments

no comment

---

## Round 0.3 · Minor Revisions

I am happy to acknowledge that the Reviewer's comments were successfully addressed (please take notice of small corrections suggested).

I still miss (I could not find) the answers to my questions raised already, which I transcribe next:

I have a major question, regarding the analysis of nuclear 28S rDNA. Being nuclear, it is expected to be diploid and therefore to harbour instances of heterozygosity. So, how was haplotyping performed/inferred?

Besides, being an RNA (and not protein) coding gene, it seems to me it makes no sense the use stop codons to filter pseudogenes.

I also share the worry about the use of a single nuclear genomic region (although I do not necessarily advise the use of ddRADseq)

I do hope the Authors can successfully address these questions.

Reviewer 4 ·

Basic reporting

The authors have addressed all of my previous comments, with the exception of one for which I am happy to accept the rebuttal. The paper is now of an excellent standard and will be of broad interest to the phylogeographic and isopod taxonomy community.

In my reading of the article I spotted some very minor errors which can be sorted out at a later stage:
49: cut "for"
439: "suggest" to "suggests"
Fig 2 caption: F seems to be the same as D- did the authors mean "MrBayes" instead of "RaxML"?

Experimental design

no further comment

Validity of the findings

The interpretations of the analyses are appropriate and the authors have explained the importance and novelty of their research.

Additional comments

no comment

---

## Round 0.4 · accepted · Accept

I think all criticisms (mine and from previous Reviewers) were successfully addressed.